# Marker-Assisted Selection for Early Maturing *E* Loci in Soybean Yielded Prospective Breeding Lines for High Latitudes of Northern Kazakhstan

**DOI:** 10.3390/biom13071146

**Published:** 2023-07-18

**Authors:** Raushan Yerzhebayeva, Svetlana Didorenko, Aigul Amangeldiyeva, Aliya Daniyarova, Shynar Mazkirat, Alyona Zinchenko, Yuri Shavrukov

**Affiliations:** 1Kazakh Research Institute of Agriculture and Plant Growing, Almaty District, Almalybak 040909, Kazakhstan; svetl_did@mail.ru (S.D.); aigul_seidinabiyeva@inbox.ru (A.A.); daniyarova85@inbox.ru (A.D.); shynarbek.mazkirat@gmail.com (S.M.); 2Breeding Station ‘Zarechnoe’, Kostanay District, Zarechnoe 111108, Kazakhstan; zinchenko.av@inbox.ru; 3College of Science and Engineering, Biological Sciences, Flinders University, Adelaide, SA 5042, Australia; yuri.shavrukov@flinders.edu.au

**Keywords:** breeding line, early maturity, *E* loci, hybrid population, marker-assisted selection (MAS), photoperiod adaptation, soybean

## Abstract

The photoperiodic sensitivity of soybean (*Glycine max* L.) is one of the limiting factors affecting plant growth and yield. At higher latitudes, early flowering and maturity with neutral reaction to photoperiods are required for adaptation of soybean plants to long-day conditions. Currently, the production and distribution of new varieties of soybeans adapted to widespread agricultural regions in northern Kazakhstan is in strong demand. Eleven soybean hybrid populations were obtained from crosses between 17 parents with four maturity groups, MG 000, 00, 0, and I. Marker-assisted selection (MAS) was assessed for suitable SSR markers and successfully applied for genes *E1*, *E3*, *E4*, and *E7*, targeting homozygous genotypes with recessive alleles. The identified and selected genotypes were propagated and tested in the conditions of 53° N latitude in the Kostanay region of northern Kazakhstan. Finally, 20 early maturing F_4_ breeding lines were identified and developed with genotypes *e1 e3 E4 e7*, *e1 E3 E4 e7*, and *e1 E3 e4 e7*, all completing their growth period within 92–102 days. These breeding lines were developed by MAS and should provide very prospective superior varieties of soybean for northern Kazakhstan through a strategy that may be very helpful to other countries with high latitudes.

## 1. Introduction

Soybean (*Glycine max* (L.) Merr.) is the world’s leading oilseed with 355.4 million tons of seeds and 58.5 million tons of oil produced across the world in 2020, more than double that of rapeseed canola oil [1]. Soybeans are a rich source of vegetable oil and protein. They are a product of multi-purpose use: food, feed, medical, and technical [2]. World soybean production has shown a strong tendency for growth every year (1994–2021), with some exceptions. Its area increased from 62.5 to 129.5 million ha in the same period [1]. Although soybean is a short-day crop with high photoperiod sensitivity, it is cultivated in a variety of climates ranging from 35° S up to 56° N [3,4,5]. Soybean has become this widespread due to breeding achievements based on the genetics of flowering, maturity, and photoperiod sensitivity [3,4,6,7]. However, each specific soybean variety has a limited latitudinal range. When a variety is transferred to another zone with a longer summer day, flowering is delayed in most varieties and the beans do not have time to ripen before the onset of cold weather [8]. For the convenience of zoning and cultivation, a classification system of 13 maturity groups (MG) has been developed and is widely used, which was created to assess the range of ecological adaptation in North America [9]. It includes groups from MG 000 to MG X. Identification of samples with earlier maturity than MG 000 in the gene pool of northern Chinese varieties and varieties from the Amur region in the Far-East of Russia led to the proposal of another early maturity rank, MG 0000. The researchers called such genotypes varieties of “high latitude cold regions” (HCR) [10].

The molecular genetic bases of soybean early flowering, early maturing, and adaptation of plants to different cultivation zones were ascribed to genetic regions designated mostly as *E* loci (Early maturity) and also a gene *J* (Juvenility). The first description and inheritance of *E* gene was made by F. V. Owen almost a century ago [11], who showed a single *E* gene with classical Mendelian segregation and linked with gray pubescence. Since that time, eleven *E* and one *J* loci have been described and studied that control the time to flowering and maturing in soybean. All these genes were perfectly summarized in a range of reviews, given the importance of soybean habit for early development [3,5,6,8,12,13,14]. Recessive alleles of the first two genes, *e1e1* and *e2e2*, control the early maturing habit, whereby the effect of the first *E1* gene was found to be much stronger [15]. Two other genes, *E3* and *E4*, were shown to be particularly sensitive to photoperiods, and plants with dominant alleles were delayed in flowering and maturing [16,17]. A smaller effect was reported for the *E5* gene in a different genetic background for time to maturity [18]. Alternatively, the *E5* gene maybe does not exist and was instead confused with the *E2* gene, as reported in a later study [19]. In contrast to previously identified *E* loci, dominant alleles of *E6* control insensitivity to photoperiod and early time to maturity in soybean [20]. Two other genes, *E7* and *E8*, were identified based on isolines of the well-known genotype of cv. Harosoy [21,22], and since that time, these isolines have been widely used in all experiments on the early maturity trait in soybean. Dominant alleles of *E9* were reported as controlling early flowering and maturity [23,24], and this phenotyping effect was similar to those in soybean plants with dominant alleles of the *E6* gene. Recent molecular analyses identified *E10* and *E11* genes, whereby soybean plants with recessive *e10e10* genotypes matured 5–10 days earlier than those with dominant *E10E10* [25]. Similarly, the role of the *E11* gene was relatively substantial, accounting for about 15% of the phenotypic variation of the early maturity trait [26]. The history of major genes controlling time to flowering and maturity concludes with the *J* gene for long juvenility (LJ) trait and delay in flowering [27]. Such LJ soybean genotypes are extremely successful in lower latitude countries such as Brazil and account for 23.7% of world soybean seed production [28]. ‘Loss of function’ mutations in soybean genotypes with recessive alleles *j* showed later flowering and maturing, which is an important trait for regions with tropical and subtropical climates [29,30].

In higher latitude countries and regions including northern Kazakhstan, four loci, *E1*, *E2*, *E3*, and *E4*, were shown to be the most important in producing soybean genotypes with early time to flowering and maturity [4,24,31,32]. Additionally, the *E7* gene was also indicated as insensitive to photoperiod and valuable for HCR [33,34].

Plant genotyping using molecular markers is one of the most rapidly developing areas in plant biology [35] and crop breeding [36], including soybean [37]. Simple sequence repeat (SSR) or microsatellite markers [38,39] are not only ‘simple’ in their structure but also in the application, where laboratories with regular equipment and tools can carry out routine molecular analyses in various crops including soybean [40]. Based on such analyses, marker-assisted selection (MAS) became popular, well-developed, inexpensive, and widely used for crop breeding [41,42]. In soybean, MAS was applied and used for many traits, including early maturity, based on targeted *E* loci [4,32,33,34].

In Kazakhstan, soybeans are cultivated mainly on irrigated fields in regions of Almaty (24.3 K ha) and Zhetysu (74.5 K ha). Currently, there is a strong demand for new soybean cultivars suitable for wide areas of northern Kazakhstan, where ecotypes have to be well adapted and insensitive to photoperiod. The Kazakh Research Institute of Agriculture and Crop Production (https://kazniizr.kz; accessed on 16 July 2023) is the main national Breeding Center in southeastern Kazakhstan, at 43° N latitude, where the largest soybean germplasm collection is kept, and most hybridizations are carried out with further evaluation and selection of progenies at F_2_–F_4_ generations [43]. Soybeans of various ripening groups, from MG 000 to MG IV are cultivated in this environment, but early maturing parents from MG 00 and MG 000 can flower over a period of only 8–10 days. Therefore, seeds are usually sown at three time-points to expand the possibility for crossings and increase the efficiency of hybridization. The selected hybrid material (F_3_-F_4_) is usually transferred to other Crop Breeding Stations located in all parts of Kazakhstan, in the south, east, north, and west of the country, depending on MG status. Selection of genotypes and further breeding programs intended for the southern area, at 42–44° N latitude, are focused on genotypes with MG II and MG III. Similarly, selection and field trials of soybean breeding lines with earlier maturity intended for northern parts, at 52–53° N, are carried out in the local Breeding Station ‘Zarechnoe’ [44,45].

In order to distribute soybean hybrid populations best suited to the different zones of Kazakhstan, the identification of the allelic diversity of *E* loci is strongly required. The breeding process for insensitivity to photoperiods is complicated because this trait can be scored only under long-day conditions, a requirement that is met through field trial testing in northern regions of Kazakhstan. To select plants insensitive to the photoperiod at the seedling stage or young plants, simple and proven DNA-based molecular markers can be used to track recessive alleles of *E* loci in segregating populations. In this regard, molecular markers and MAS were introduced to assess insensitivity to photoperiods, to identify and select soybean progenies from hybrid populations suitable for distribution and cultivation in Kazakh northern regions.

The aims of this study included: (1) Assessment and application of MAS for four *E* loci: *E1*, *E3*, *E4*, and *E7*, for insensitivity to photoperiods in soybean hybrid populations, progenies, and developed breeding lines and (2) Confirmation of MAS-based developed soybean breeding lines with high yield in field trial tests at high latitudes (52–53° N) in the northern region of Kazakhstan.

## 2. Materials and Methods

### 2.1. Plant Material

The plant research accessions included 17 soybean parental germplasm with four maturity groups MG 000, 00, 0, and I, as well as 11 of their hybrid populations (F_2_–F_3_). The studied soybean accessions are listed and described in Section 3 below. Additionally, positive and negative controls, isogenic lines with well-known recessive and dominant alleles of the four genes, *E1*, *E7*, *E3*, and *E4*, were used. They included: (1) Harosoy OT94-47 with fully recessive genotype (*e1 e3 e4 e7*); (2) Harosoy OT89-5 with dominant allele *E7* (*e1 e3 e4 E7*); (3) cv. Harosoy, original accession with recessive allele *e1* (*e1 E3 E4 E7*), and (4) Harosoy OT93-26 with recessive allele *e3* (*E1 e3 E4 E7*). For the Kostanay region (northern Kazakhstan), the two soybean cultivars, Ivushka (Kazakhstan) and SIBNIIK-315 (Russia) were approved and used in this area as local standards. As international standards, the following soybean varieties were used: Maple Amber, Canada (Registration No. 2111, Canadian Variety Registration, Toronto, 1981), Fiskeby III, Sweden (Accession No. PI 438471), and Maleta, Russia (Registration No. 11115, St.-Petersburg, VIR, 2007). All used standards showed an early maturing habit during preliminary testing in the Kostanay region. Both local and international standards (or checks) were used in field trials for comparison with studied accessions and breeding lines.

Seeds were obtained from the local germplasm collection of the Kazakh Research Institute of Agriculture and Plant Growing (KRIAPG), Almaty region, Kazakhstan, and also kindly provided by the Breeding Company ’Soya-Sever’, Minsk region, Belarus, and Australian Grains Genbank AGG, Horsham, Australia.

### 2.2. Hybridization

To make crosses, prior to the hybridization, the color of the flower corolla in parental plants was taken into account as a marker trait. White-flowered accessions were used as maternal plants only with recessive genotype *w1w1*, determining the white color of flowers [46]. For paternal parents, plants were selected with the purple color of the flower, encoded by the dominant allele of the gene *W1w1*. The identification of true F_1_ hybrids was carried out in the flowering stage of hybrids for the purple color of the flowers.

Hybridization was carried out in the research field in soybean developmental stages R1-R2, beginning or early flowering. It started from days 3 to 5 of R1 stage, between 7:00 and 11:00 AM. Unopened flowers with closed sepals (corolla petals not yet visible) were selected for use as female flowers. All other flowers and buds on the node were removed. The procedure was carried out without emasculation of anthers. Briefly, after a vertical incision of sepals and petals of the corolla with a thin sharp needle, only three anthers located directly under the stigma were removed from the flower to make the stigma clearly visible [47]. Pollination was conducted immediately using fresh pollen from unopened flowers with partially open sepals. Pollen was carefully applied with a needle to the stigma of the pistil. Upon completion of this procedure, the flower was tied with a thin bright cotton thread for labeling [48].

### 2.3. DNA Extraction, PCR, Molecular Markers, and Marker-Assisted Selection (MAS)

For MAS, parents and all plants of each hybrid soybean population were numbered and labeled. After plants had grown three pairs of true leaves, one leaf from each individual plant was collected for sampling in 10 mL plastic tubes and stored at −20 °C. DNA was extracted from the leaves of each labeled plant using the CTAB method [49]. DNA quality was assessed in agarose gel electrophoresis and quantified using a nanodrop spectrophotometer (Thermo Fisher, Waltham, MA, USA). Labeled plants continued their growth and development.

SSR markers with known and verified associations with *E* loci were used in this study [30]. Molecular markers Satt557 and Satt365 were used for targeting the *E1* gene, marker Satt229 for *E3* gene, markers Satt100 and Satt319 for *E7* gene, and CAPS marker (cleaved amplified polymorphic sequences) PhyA2 was used for *E4* gene analysis [50]. In the current study, PhyA2 marker was polymorphic after PCR amplification, and did not require the use of endonuclease enzymes. Sequences of primers for DNA markers and amplification conditions are presented in Appendix A.

PCR analysis was carried out in an Eppendorf MasterCycler (Berzdorf, Germany). The composition of the reaction mixture for PCR analysis in a total reaction volume of 15 µL was as follows: 100 ng of genomic DNA, 1 × PCR buffer, 2.5 mM MgCI_2_, 200 µM of each dNTP, 0.5 µM of each primer, 1 µg of BSA, and 0.5 Units of Taq polymerase (Biosan, Novosibirsk, Russia).

Fragment separation and size detection was carried out by electrophoresis of amplified products in 8% polyacrylamide gel (Sigma-Merck, China), stained with ethidium bromide, and visualized and imaged using a UV Transilluminator Quantum ST4 (Vilber-Lourmat, Marne-la-Vallée, France). DNA marker Step50 (BiolabMix, Novosibirsk, Russia) was used for molecular weight identification of PCR products.

### 2.4. Research Field Experiments and Field Trials

Experiments with parents, as well as hybridization and selection of individuals with desired early maturity traits in F_1_-F_3_ generations were carried out at KRIAPG in 2017–2021. The research field trial was located in a hill zone 740 m above sea level, geographical location 43°15′ N and 76°54′ E, in the Almaty District of southeastern Kazakhstan. According to Köppen classification [51], the climate of the Almaty region is ‘Dfa’, a continental climate with hot summers. The average annual temperature is 6.5 °C and the amount of precipitation is 891 mm [52]. As a result of the selection based on DNA markers, 103 breeding lines of the F_4_ generation were transferred to the breeding station in northern Kazakhstan.

Further field trials were carried out in northern Kazakhstan based on the analysis of a set of traits with 103 breeding lines propagated from soybean individuals with identified desired allelic combinations of *E* loci. These tests were conducted in 2022 at Breeding Station ‘Zarechnoe’, Kostanay District, with geographical location 53°12′ N and 63°37′ E. The climate of Kostanay region is ‘Dfb’, a continental climate with cold winters and warm summers [51]. The average annual temperature is 4.2 °C and the amount of precipitation is 422 mm [52].

The classical pedigree method was applied for the analysis and selection of progenies from soybean hybrids [53]. Seeds of each F_1_ hybrid combination were sown separately in plots between similar plots of both parents for their clear comparisons. Color of the corolla flower in each individual plant of F_1_ hybrids was assessed and only plants with colored corolla flowers were designated as ‘true’ hybrids while ‘escaped’ plants with white colored corolla were discarded. Analysis and harvesting of F_1_ plants was conducted strictly individually as progenitors of progenies and further breeding lines. Therefore, starting from F_2_ and in further generations, seeds were sown separately by progenies originating from a single F_1_ plant. Before sowing, the seeds were checked and selected for their similarity, lack of pigmentation, wrinkling, and disease damage. The plants of the studied F_2_–F_3_ soybean hybrid populations were grown in 2020–2021 on field plots, in a single row one meter in length with 25 seeds per plot sown at 4 cm depth.

During the growing period, phenological observations of the growth and development of soybean plants were carried out according to the methodology described earlier [54]. After complete analysis of all studied traits, the association between early maturity and high yield was determined; based on this, the most promising individual plants were identified and selected. The assessment of productivity elements was carried out according to the methodological recommendations [55]. Thousand seeds weight (TSW) was measured according to the Governmental Standard, Kazakhstan (No. 12042-80) [56].

### 2.5. Statistical Treatment

Statistical treatment of the received data was carried out with the Program R, version 4.1.2 R [57]. In this study, a standard nonparametric Pearson’s Chi-squared test was performed to identify the association of the frequencies of the studied factors and received results. The data for group maturity and allelic combinations of four studied *E* loci were analyzed for 667 soybean plants tested in experiments in the Almaty region of southern Kazakhstan.

One-way analysis of variance ANOVA was applied to identify statistical differences between genotypes and their maturity time in all hybrid populations and developed breeding lines. Multiple pairwise comparisons of means of growing duration and flowering time of data between groups with different allelic combinations of *E* loci in soybean progenies and breeding lines compared to standard varieties were performed using a Post Hoc test Tukey’s HSD (Honestly Significant Difference), to assess the significance of differences between pairs of group means. In this study 104 soybean breeding lines were analyzed.

## 3. Results

### 3.1. Selection of Parents for Crosses and Hybrid Analysis

The soybean germplasm collection used in this study was analyzed both for genotyping of the allele diversity in *E* loci, and for phenotype assessments for maturity groups, growth type, weight of 1000 seeds, and seed yield per m^2^ in the research field. The summary results for both genotyping and phenotyping in research fields both in the Almaty and Kostanay Districts of Kazakhstan are shown in Table 1. The seed yield per plot, TSW, and maturity groups were assessed in field trials in 2019–2021. Two varieties, Birlik KV and Zara, with large white flowers were selected as maternal parents while another 15 varieties with colored flowers were chosen for crossings as paternal parents. For soybean hybridization, only unopened flowers are used, and their size is very small, only 4–5 mm in plants of MG 00 and MG 000. The presence of larger flowers with a white corolla in maternal plants is a very important trait, increasing the efficiency of hybridization. Soybean varieties from maturity groups MG 000–I were selected for paternal parents, that were suitable to grow in high latitude conditions, such as 50–53° N, with high yield, non-dehiscent pods, and having valuable recessive alleles of *E* loci for photoperiod insensitivity. Additionally, soybean varieties with dominant alleles of *E* genes were included in the hybridization: Pamyat YKG, as a high-yield genetic resource; Grignon 5, for drought tolerance, and Soer 345, as a non-shattering pod germplasm accession.

Soybean hybridization represents a very delicate process and is almost similar to a ‘microsurgical operation’. Firstly, plants are very short (20–30 cm) at the beginning of flowering, especially in MG 000–MG 0 early maturing groups. Additionally, soybean accessions, insensitive to photoperiod and adapted to northern latitudes, showed different reactions when they grew in southern Kazakhstan. These included earlier flowering, a shorter growth period, and smaller flowers and floral organs compared to local soybean varieties with high sensitivity to the photoperiod.

Additionally, semi-sterile anthers with a partial absence of pollen grains were noticed and recorded in Maple Amber, Fiskeby III, and Maleta cultivars, used as international standards. In total, 15 hybrid combinations were produced and studied after manual hybridization of 30 flowers in each cross. The success rate of hybridization varied from 0 to 23.3%, depending on the hybrid combination. In three hybrids, Birlik KV × Altom, Zara × Fiskeby III, and Zara × Maple Amber, all hybridizations were unsuccessful and no seeds were obtained. In 2019, the ‘true F_1_ hybrids’ were identified and selected for further experiments. In total, 38 original F_1_ hybrid plants were selected from 77 plants (49%) in 11 crosses. These F_1_ hybrid plants represented progenitors of breeding lines in further generations of study. The efficiency of hybridization is presented in Appendix A.

### 3.2. Molecular Markers and MAS for Four Genes, E1, E3, E4, and E7, Controlling Photoperiod Sensitivity in F_2_–F_3_ Hybrids of Soybean

All parents of the produced and studied hybrids were tested for four *E* genes, *E1*, *E3*, *E4*, and *E7*, prior to the start of further experiments, and their genotyping results are presented in Table 1. Two maternal parents are shown in Table 1A, while paternal parents are arranged in two groups, which were crossed with cv. Birlik KV (Table 1B) and with cv. Zara (Table 1C).

For 11 hybrid populations, seeds of 38 F_2_ families originating from the corresponding F_1_ progenitor plants were identified in 2020 for the allelic diversity of *E1*, *E3*, *E4*, and *E7* genes. Based on DNA marker genotyping of each individual plant, homozygous recessive alleles of *E* genes were assessed and identified. In hybrid populations heterozygous for any of the four studied *E* genes, individual plants were self-pollinated in 2021, and homozygous F_3_ genotypes were identified and selected for further analyses. In total, 667 individual plants from hybrid populations were studied with molecular markers and MAS for four genes, *E1*, *E3*, *E4*, and *E7*. The genotyping results of each of these *E* genes are presented separately in the following sub-sections.

#### 3.2.1. Identification of *E1* Allelic Variation

Genotyping with the first marker, Satt365, was recorded using the two fragments, A and B. The first fragment A corresponded to genotype *E1E1* with dominant alleles, and it was found in soybean line Harosoy OT93-26, which was used as a negative control for the early maturity trait. In contrast, the second fragment B was found in line Harosoy OT94-47, and was used as a positive control for recessive allele *e1e1*. The second marker, Satt557, was typically used for the confirmation of the first marker, Satt365, in all studied hybrid combinations. However, one crossing combination, Birlik KV × Grignon 5, was the exception because 11 out of 60 plants showed conflicting genotyping results in the identification of dominant and recessive alleles of the *E1* gene. In this combination, MAS was carried out according to the results of the first marker, Satt365. Based on the PCR analysis of individual plants from overall F_2_–F_3_ hybrid populations using two markers, Satt365 and Satt557, 311 identified plants had valuable genotypes with homozygous recessive alleles *e1e1*. Figure 1 shows two examples of individual plant genotyping for alleles of the *E1* gene using marker Satt557 with all variants of homo- and heterozygotes (Figure 1a) and selected homozygote genotypes *e1e1* and *E1E1* (Figure 1b).

Similar genotypes *e1e1* were identified in the following hybrid populations: Birlik KV (*e1*) × Rana (*E1*); Birlik KV (*e1*) × Pripyat (*e1*); Birlik KV (*e1*) × Grignon 5 (*E1*); Birlik KV (*e1*) × Pamyat YKG (*E1*); Birlik KV (*e1*) × Toury (*e1*); Birlik KV (*e1*) × Soer 345 (*E1*); Zara (*E1*) × Maleta (*e1*), and Zara (*E1*) × Ustya (*e1*) (Table 2).

#### 3.2.2. Identification of *E3* Allelic Variation

Identification of plants in soybean segregating generations for gene *E3* was conducted with the application of marker Satt229 (Figure 2a). Two fragments, A and B, were amplified using the Satt229 marker, and these fragments were identified as corresponding to the negative control, cv. Harosoy (*E3E3*) and the positive control, Harosoy line OT89-5 (*e3e3*), respectively. Similar DNA analysis of plants was also carried out in two hybrid populations, Zara (*E3*) × Maleta (*e3*) and Birlik KV (*E3*) × Rana (*e3*) (Table 2). The recessive allele *e3* was identified in these hybrids, including 40 plants in Birlik KV × Rana and 17 plants in Zara × Maleta hybrid populations. Therefore, 57 out of 216 individual plants were identified as homozygotes of recessive allele *e3*, which is responsible for insensitivity to photoperiod. In all other hybrids, parental plants had only the dominant allele of the *E3* gene. These hybrids were excluded from further analysis due to the absence of segregation for the *E3* gene.

#### 3.2.3. Identification of *E4* Allelic Variation

For *E4* gene identification, soybean plants in hybrid segregating populations were studied using molecular marker PhyA2. It included analysis of two PCRs with one common Forward primer (PhyA2-For) and with one of two Reverse primers (PhyA2-Rev/e4 or PhyA2-Rev/E4). The use of molecular marker PhyA2 resulted in two groups of plants with contrasting genotypes of the *E4* gene. The first group with 65 plants showed the amplified fragment of 837 bp, which corresponds to the favorable recessive allele *e4* with identical frame size in the positive control, Harosoy OT94-47 (*e4e4*). The second group included 204 plants with a PCR fragment of 1229 bp, corresponding to the dominant allele *E4*, confirmed with the negative control, cv. Harosoy (*E4E4*) (Figure 2b). Additionally, 22 heterozygous plants *E4e4* were also recorded in this experiment. It is important to note that this analysis was carried out only in five hybrids, where paternal parents had the recessive allele *e4*: Birlik KV × Pripyat; Zara × Bara; Zara × Maleta; Zara × Ustya, and Zara × Jhony.

#### 3.2.4. Identification of *E7* Allelic Variation

To identify allelic variation of the *E7* gene, two microsatellite markers, Satt100 and Satt319, were employed. In this study, the Satt100 marker showed high polymorphism. Four fragments were identified during the analysis of parental forms and hybrid populations: (A) 150 bp; (B) 145 bp; (C) 180 bp, and (D) 120 bp. Fragments A (150 bp) and B (145 bp) correspond to dominant and recessive alleles, *E7* and *e7*, with identical lengths of amplified products recorded from the negative (Harosoy *e3 e4 E7*) and positive (Harosoy *e3 e4 e7*) controls, respectively. Amplified fragment C (120 bp), designated as allele *C*, was identified only in cv. Rana and in its hybrid populations. In contrast, fragment D (180 bp) was found in four varieties, Grignon 5, Pamyat YGK, Soer 345, and Zara, and in their hybrid populations, and was designated as allele *D* (Figure 3a).

The correct identification of the recessive allele *B* (*e7*) and applied MAS for plants with homozygote genotype *e7e7* in this analysis was verified and confirmed using the second marker Satt319. All other plants with *A*, *C*, and *D* alleles based on genotyping with marker Satt100, produced amplified products with marker Satt319 identical to those in the negative control Harosoy OT89-5 with dominant genotype *E7E7* (Figure 3b). Therefore, in this study, they were identified as plants with dominant allele *E7*.

Based on PCR analysis for two markers, Satt100 and Satt319, 298 individual plants with the homozygous genotype *e7e7* were identified and selected for further analysis for insensitivity to photoperiod. It is important to note that *E7* and *E1* genes were found linked in a genetic fragment in almost all studied genotypes.

### 3.3. Association Analysis for Genotypes of Four E Genes and Maturity Groups in Conditions of Southern Kazakhstan

Based on the identification of the allele composition in all 667 studied plants from 11 hybrid populations, plants were arranged into 10 groups of genotypes: (1) *e1 e3 E4 E7*; (2) *e1 E3 e4 e7*; (3) *E1 E3 e4 E7*; (4) *E1 E3 e4 e7*; (5) *E1 E3 E4 E7*; (6) *e1 e3 E4 e7*; (7) *e1 E3 E4 e7*; (8) *E1 e3 E4 E7*; (9) *e1 E3 E4 E7*, and (10) *E1 E3 E4 e7*. The significant differences for associations were assessed between results of allelic variations of *E* genes in 10 groups of genotypes and data of the maturity groups of progenies received from all 667 tested plants of F_2_–F_3_ hybrid populations grown in the Almaty region, southern Kazakhstan.

The results of the association analysis (Pearson’s Chi-squared test) are shown in Figure 4. Blue dots indicate positive scale values of associations and burgundy dots show negative scale values of associations. The dot size and color intensity reflect the direct value of each association calculated mathematically using Pearson’s Chi-square test, that is, the biggest dot with the most intense color indicates the strongest association while a lower association is indicated by a smaller dot with milder color intensity.

The highest association, at the level of +4.1, was found between genotypes *e1 e3 E4 e7* and ultra-short maturing group MG 00. In the second early maturity group MG 0, the highest level of association (+4.45) was identified with genotype *E1 E3 e4 e7*. In contrast, genotype *E1 E3 E4 E7* showed the lowest association level at −3.97 in ultra-short maturity group MG 00. These Chi-square association values indicate that plants with *E1 E3 E4 E7* genotype cannot be present in the early maturity group (Figure 4).

### 3.4. Plant Analysis and Field Trials in Southern Kazakhstan

The analysis of plant growth and productivity was carried out with progenies of 667 plants originated from 11 hybrid populations in comparison with parents and Harosoy isolines in the conditions of southern Kazakhstan. The average growth period of Harosoy isolines included 96 days for OT94-47 (*e1 e3 e4 e7*), 99 days for OT89-5 (*e1 e3 e4 E7*), and 118 days for cv. Harosoy (*e1 E3 E4 E7*). The average seed weight per plant (SWP) in the isolines was 5.4 g in OT94-47, 7.5 g in OT89-5, and 20.6 g in cv. Harosoy.

For individual plants in hybrid populations, the average growth period ranged from 91 to 139 days. The SWP in the studied hybrid populations varied from 7.4 g (Birlik KV × Pamyat YGK) to 24.1 g (Zara × Jhony). The TSW was from 158.52 g (Zara × Ustya) to 206.1 g (Zara × Bara). The average values of seed yield components in plants from 11 hybrid populations are presented in Appendix A. Based on the results of this analysis, 103 breeding lines with high SWP and TSW with six genotypes were selected for testing in the northern regions of Kazakhstan.

### 3.5. Plant Analysis and Field Trials in Northern Kazakhstan

After identification of homozygous genotypes with recessive alleles of *E* genes, 103 high-yielding soybean breeding lines with six groups of genotypes were selected for analysis in field trials in the Kostanay region (53°21′ N). These six selected groups of genotypes were as follows: (1) *e1 e3 E4 e7*; (2) *e1 E3 E4 e7*; (3) *E1 E3 e4 E7*; (4) *E1 e3 E4 E7*; (5) *e1 E3 e4 e7*, and (6) *E1 E3 E4 E7*. The list of the 103 selected breeding lines, their allelic polymorphism in *E* genes, and records for flowering and maturation time in field trials in the Kostanay region are presented in Appendix A. The studied genotypes started flowering at different times, ranging from 33 days after seedling emergence (Line P-6/3-2, from hybrid Birlik KV × Toury, with genotype *e1 E3 E4 e7*) to 58 days (Line P-18/5-6, from hybrid Zara × Soer 3, with genotype *E1 E3 E4 E7*).

In particular, 13 breeding lines were noted with a flowering period of 33–38 days, originating from hybrids Birlik KV × Rana, Birlik KV × Toury, Birlik KV × Pripyat, Zara × Maleta, and Zara × Jhony, flowering earlier and similar to international standard varieties Maleta (37 days) and Maple Amber (38 days). The highest frequency of early flowering breeding lines for 33–39 days (52.9%) was found in the group with genotype *e1 e3 E4 e7*, while late flowering plants (50–58 days) mainly belonged to two groups of genotypes, *E1 E3 e4 E7* (80%) and *E1 E3 E4 E7* (35.4%) (Figure 5).

The analysis of variance in flowering time of soybean breeding lines showed significant differences at a very high level of significance, *p* < 0.001 (Figure 6). Pairwise comparisons of the mean values of flowering time in the studied groups of genotypes compared to local and international standards indicated for significant differences between early flowering genotypes *e1 e3 E4 e7*, in standards Maleta, Maple Amber, SIBNIIK-315, and Ivushka, and late flowering genotypes, *E1 E3 e4 E7* and *E1 E3 E4 E7* (Table 3).

Comparison of the studied breeding lines with the corresponding Harosoy isolines with the same genotypes showed no significant differences in flowering time. Isoline OT94-47 (*e1 e3 e4 e7*) started flowering 41 days after seedling emergence, line OT89-5 (*e1 e3 e4 E7*) after 43 days, and cv. Harosoy (*e1 E3 E4 E7*) delayed flowering up to 46 days (Table 3).

Fully mature breeding lines in the Kostanay region began to be recorded on September 3, and recording finished on 28 September 2022. In the environment of northern Kazakhstan, 63 out of 103 studied breeding lines reached full maturity at harvest. The length of their growth period ranged from 92 to 121 days. In contrast, 40 soybean breeding lines failed to produce seeds at harvest, and their maturation period was presumably about 130 days or more. The highest frequency of genotype occurrence with early maturation of 92–99 days was observed in the group with genotype *e1 e3 E4 e7* (41.2%). Non-matured plants with delayed seed production up to 130 or more days were found with a higher frequency in two groups of genotypes, *E1 E3 e4 E7* (80%) and *E1 E3 E4 E7* (64.5%) (Figure 7).

The ANOVA analysis of the growing duration of six groups of genotypes with 104 soybean breeding lines showed significant differences between the groups at *p* < 0.001 probability (Figure 8). Significant differences were found between one genotype *e1 e3 E4 e7* and two other contrasting genotypes, *E1 E3 e4 E7* and *E1 E3 E4 E7* (Table 3).

According to the results from field trials, 20 promising and early maturing breeding lines were identified, including 14 lines from hybrid Birlik KV × Rana, two lines from Birlik KV × Grignon 5, and one line each from four hybrids, Birlik KV × Pripyt, Birlik KV × Soer 345, Birlik KV × Toury, and Zara × Maleta, with a growth period of 92–102 days. Such a relatively short growing period was similar to those in local standards, SIBNIIK-315 (95 days) and Ivushka (102 days) as well as in international standards, cvs. Maleta (98 days) and Maple Amber (102 days). Examples of breeding lines with contrasting time to maturity, selected from hybrid Birlik KV × Rana at harvest in the fully mature stage, are shown in Figure 9. About 70% of early maturing soybean lines were selected from this segregating hybrid population.

Among 20 selected breeding lines, SWP varied from 12.1 g to 18.6 g. The highest yields were 18.6 g in line P-9/3-6 (Birlik KV × Soer 345), 18.2 g in line 2/5-4 (Birlik KV × Pripyat), and 17.9 g in line P-1/7-4 (Birlik KV × Rana). The TSW varied from 124 g to 156 g, and the largest were 156 g in lines P-2/5-4 (Birlik KV × Pripyat), 149 g in line P-3/3-1 (Birlik KV × Gignon 5), and 148 g in line P-1/6-4 (Birlik KV × Rana). The results for seed yield, TSW, and time to maturity in 20 selected breeding lines grown in field trials in Kostanay region are presented in Appendix A.

Twenty promising and early maturing soybean breeding lines were represented by the following genotypes: 11 lines (55%), *e1 e3 E4 e7*; three lines (15%), *e1 E3 E4 e7*; two lines (10%), *E1 e3 E4 E7*; two lines (10%), *e1 E3 e4 e7*, and two lines (10%), genotype *E1 E3 E4 E7*.

In our study, it was found that lines with the same genotype for the *E1*, *E3*, *E4*, and *E7* genes differed in maturation time. This suggests the contribution of other *E* loci to these differences and a consequence of the effect of temperature but there was no evidence to support the hypothesis in the current study. A comparison of the time to flowering and maturity showed that 25 out of 34 breeding lines (75.5%) had a delay to the start of flowering after 45–58 days and, therefore, subsequently did not reach full maturity. These 25 lines with delayed flowering were presented by four genotypes: *E1 E3 E4 E7* (14 lines), *e1 E3 E4 e7* (6 lines), *E1 E3 e4 E7* (3 lines), and *E1 e3 E4 E7* (2 lines).

## 4. Discussion

The distribution of soybean crops to regions with a cooler climate is in strong demand, with seed production used for food, feed, oil, and industrial purposes in countries of both hemispheres. The process of soybean distribution is directly related to two major and linked factors, a shorter growth period and insensitivity of plants to longer day-length [13,31]. Perfect soybean cultivars selected in southern countries with lower latitudes are non-adaptive and fail to produce high seed yield, often producing no seeds at all in northern regions with higher latitudes. The genetic control of photoperiodic response of plants to day-length is one of the major causes and limitations for the distribution of soybean crops to higher latitude areas [5]. This also applies to Kazakhstan, a Central Asian country, where very high-yield soybean cultivars are produced and grown in the southern part at 43° N, but more adapted soybean genotypes with a neutral reaction to photoperiod are required to grow in the northern part with a higher latitude at 50–53° N [35,36,37].

In the presented study, four *E* loci, *E1*, *E3*, *E4*, and *E7*, were analyzed out of 11 known, with recessive alleles controlling early flowering and maturing of soybean plants. Eleven out of fifteen hybrid populations with 103 out of 667 breeding lines in total were selected for testing in the Kostanay region at 53° N latitude. The importance of these *E* loci was indicated earlier, whereby alleles of *E1–E4* genes explained 62–66% of variation in time to flowering [32] while soybean plants with dominant alleles *E7* resulted in a relatively mild delay of 4–5 days in flowering time [34].

Recently, several CAPS markers were developed to analyze genetic polymorphism and allele identification in *E1–E4* loci [32,58,59]. For the *E1* gene, derived CAPS marker (dCAPS) TI, based on the digestion of the amplified PCR products with the restriction enzyme *Hinf*I was successfully applied and a series with three recessive alleles were identified: *e1-as*, *e1-fs*, and *e1-nl* [58]. This result with the dCAPS marker was excellent and showed wider genetic polymorphism. However, in general, CAPS and dCAPS methods require a significantly longer time investment due to the requirement of two consequent steps, and it is more expensive, depending on the cost of individual restriction enzymes.

In the presented study, we used only SSR markers and one modified CAPS marker without digestion with a restriction enzyme, which can be considered as just an allele-specific PCR marker. In general, SSR markers represent a simple, accurate, and inexpensive method of soybean plant genotyping, and it was successfully applied for four studied *E* loci, similar to those published earlier [33,34]. However, compared to the CAPS method, SSR markers can distinguish between dominant and recessive alleles but not among recessive alleles in each *E* gene. Despite this disadvantage, SSR markers were successfully used in our study and all parents and progeny plants were genotyped at the early stage of plant development as an important step for MAS.

Based on the presented results, *e1 e3 E4 e7* was identified as the best soybean genotype with the most neutral reaction to day-length with the earliest start of flowering and maturing in the high latitude conditions of Kostanay (Table 3, Figure 6 and Figure 7). These results were similar to those published earlier [4], where one of the combinations carrying the dominant allele *E4* and recessive alleles in other *E* loci showed the earliest time to maturity in northeast China [4]. Therefore, the combination of three homozygotes, *e1e1 e3e3 e7e7*, can play a very important role and bring about the relatively quick development of soybean plants for such conditions. Interestingly, the presence of dominant allele ‘*E4_*’ (*E4E4* or *E4e4*) in these genotypes did not significantly change the plant reaction to day-length. These results showing a minor-to-moderate effect of the *E4* gene for maturity time were similar to those reported in soybean from European accessions [60], but differ from a previously published report from Japan [50]. However, different combinations of other alleles of *E* loci and genetic background could explain such contradictions.

The next three genotypes (*e1 E3 E4 e7*, *e1 E3 e4 e7*, and *E1 e3 E4 E7*) represent another group, intermediate for time to flowering and maturity (Table 3) while their variability was quite high (Figure 7), with the only exception being genotype *E1 e3 E4 E7* with a relatively narrow variability in time to flowering (Figure 6). In the first genotype, *e1 E3 E4 e7*, it is understandable that the presence of dominant alleles in the gene, ‘*E4_*’, resulted in a slight delay in flowering and time to maturity, and these results were similar to those in analyses of Chinese soybean cultivars [4,31].

The most intriguing situation occurred in two other genotypes, *e1 E3 e4 e7* and *E1 e3 E4 E7*, where the alleles of four studied *E* loci are perfect opposites—from recessive to dominant alleles in *e1 e4 e7* genotypes and vice versa, from dominant to recessive alleles in *E3* genotypes. There is no significant difference between these two genotypes with intermediate time to flowering and maturity, with only higher variability in flowering time in *e1 E3 e4 e7* genotypes (Figure 6), while genotype *E1 e3 E4 E7* was more diverse in maturity time (Figure 8). High variability and possible additional genetic factors involved in genetic control of time to maturity were also shown earlier in a study of European soybean cultivars [61]. 

The last group of genotypes with dominant alleles in all or most studied *E* loci, *E1 E3 E4 E7* and *E1 E3 e4 E7*, had a late time to flowering and maturity (Table 3, Figure 6 and Figure 8). The difference in average flowering and maturity time between these two genotypes accounted for 2.7 days and it was statistically non-significant (Table 3). These genotypes presented a lower value and interest for potential distribution in northern regions with higher latitudes, similar to results published earlier [4].

During MAS application among 667 soybean breeding lines, only 103 of them were selected and tested in the Kostanay region. In field trial tests, 20 breeding lines were identified as early maturing and promising based on their agronomic performance, where the majority of genotypes were *e1 e3 E4 e7*. Similar results were reported in Chinese cultivars distributed and moved from lower to higher latitudes (from 40° N to 53° N), where genotypes *e1 e3 E4* (without analysis of *E7e7* locus) accounted for about 36.6% of presence in local varieties [4,31].

Therefore, soybean accessions, germplasms, hybrid progenies, and breeding lines with *e1 e3 E4 e7* genotypes must be the main priority where there is interest and demand for prospective seed yield production in any regions with higher latitudes. These genotypes show the shortest time to flowering and maturity, and their genotyping can be easily assessed by simply using the developed and verified molecular markers presented in this study.

## 5. Conclusions

Marker-assisted selection (MAS) and application of molecular markers have improved breeding efficiency and the use of soybean germplasm. Based on MAS, promising breeding lines with genotypes *e1 e3 E4 e7*, *e1 E3 E4 e7*, and *e1 E3 e4 e7* were identified showing earlier maturity in the environment of northern Kazakhstan at 53° N latitude. Out of six studied genotypes of soybean breeding lines, the most promising allele combination of *E* genes controlling the insensitivity to photoperiod was *e1 e3 E4 e7*. Breeding lines with this genotype started flowering and completed the development of mature seeds significantly earlier compared to genotypes with mostly dominant alleles: *E1 E3 e4 E7* and *E1 E3 E4 E7*. Based on the results of field trial tests in the Kostanay region, 20 promising and prospective early maturing soybean breeding lines were identified with a short growth period of 92–102 days. This habit of plant development was similar to those in local standards, SIBNIIK-315 (95 days) and Ivushka (102 days), as well as in international standards, cvs. Maleta (98 days) and Maple Amber (102 days). The identified and tested selected breeding lines can be potentially very successful in developing superior performing soybean varieties for northern regions of Kazakhstan and other countries with similar climates at high latitudes.

## Figures and Tables

**Figure 1 biomolecules-13-01146-f001:**
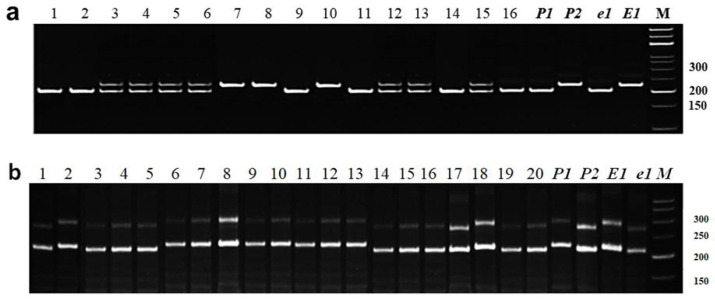
Identification of soybean plants for *E1* alleles using marker Satt557 and separation of PCR products in 8% polyacrylamide gel. (**a**) Sixteen individual plants from F_2_ hybrid Birlik KV × Rana; and (**b**) 20 individual plants from F_3_ hybrid Birlik KV × Grignon 5. Both parents, P_1_ and P_2_ (maternal and paternal, respectively), known recessive genotype, line OT94-47 (*e1e1*), and dominant genotype, line OT93-26 (*E1E1*), were included for comparison. M—molecular weight ladder Step50.

**Figure 2 biomolecules-13-01146-f002:**
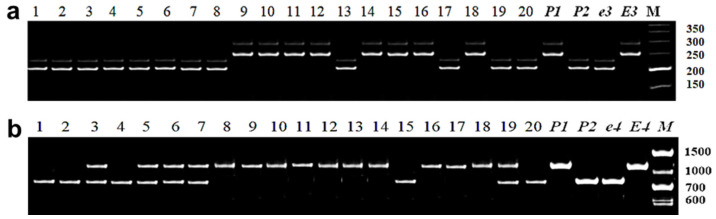
Identification of soybean plants. (**a**) *E3* alleles using marker Satt229, 20 individual plants from F_2_ hybrid Birlik KV × Rana. Both parents, P_1_ and P_2_ (maternal and paternal, respectively), known recessive genotype, line OT89-5 (*e3e3*), and dominant genotype, cv. Harosoy (*E3E3*), were included for comparison; (**b**) *E4* alleles using marker PhyA2, 20 individual plants from F_2_ hybrid Zara × Maleta. Both parents, P_1_ and P_2_ (maternal and paternal, respectively), known recessive genotype, line OT94-47 (*e4e4*), and dominant genotype, cv. Harosoy (*E4E4*), were included for comparison. PCR products were separated in 8% polyacrylamide gel. M—molecular weight ladder Step50.

**Figure 3 biomolecules-13-01146-f003:**
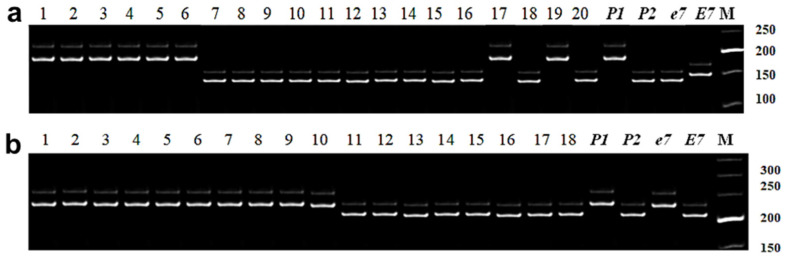
Identification of soybean plants with *E7* alleles using marker Satt100 with 20 individual plants from F_2_ hybrid Zara × Maleta (**a**), and with marker Satt319 with 18 individual plants from F_2_ hybrid Birlik KV × Pamyat YGK (**b**). Both parents, P_1_ and P_2_, in each hybrid, known recessive genotype, line OT94-47 (*e7e7*), and dominant genotype, cv. Harosoy (*E7E7*), were included for comparison. PCR products were separated in 8% polyacrylamide gel. M—molecular weight ladder Step50.

**Figure 4 biomolecules-13-01146-f004:**
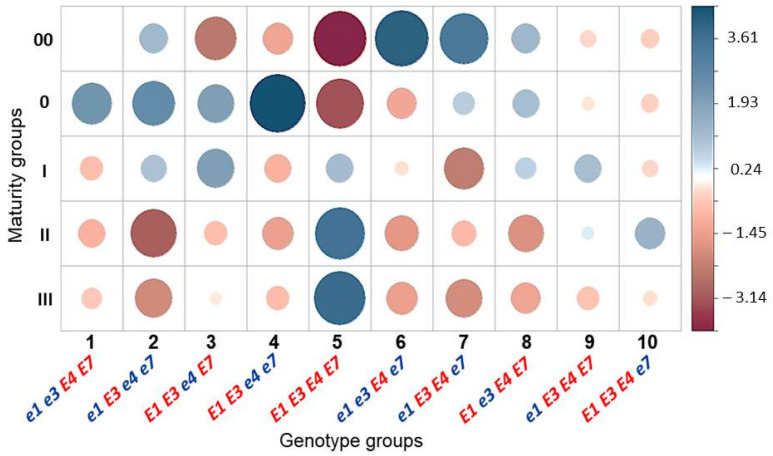
Pearson’s Chi-squared values for association analysis between allelic variation in groups of genotypes and maturity groups, and all studied soybean hybrid populations, generations F_2_–F_3_, grown in the Almaty region, southern Kazakhstan. Data: Chi-squared = 181.53; df = 36, *p* < 0.001.

**Figure 5 biomolecules-13-01146-f005:**
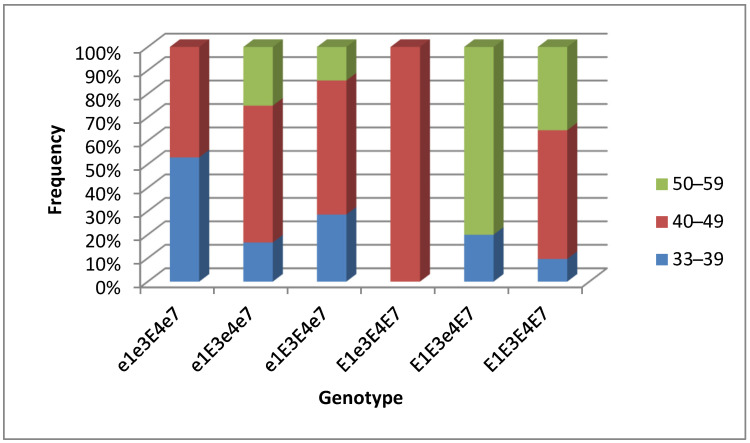
The frequency of occurrence of different periods (number of days) from seedling emergence to flowering in soybean genotypes from hybrid populations, F_4_ generation, grown in the Kostanay region, northern Kazakhstan.

**Figure 6 biomolecules-13-01146-f006:**
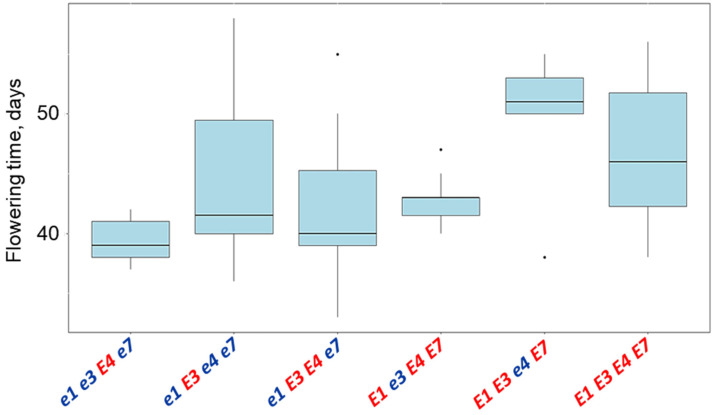
Interval plots of variance analysis for the flowering time duration of soybean breeding lines tested in the Kostanay region depending on their allelic composition of four *E* genes. Data: ANOVA; df = 6, F = 6.3, *p* < 0.001. Isolated dots represent extreme values exceeding the general variability level (0.99) of the received data. These dots are generated by R computer software and reflected automatically in the value of average and variability.

**Figure 7 biomolecules-13-01146-f007:**
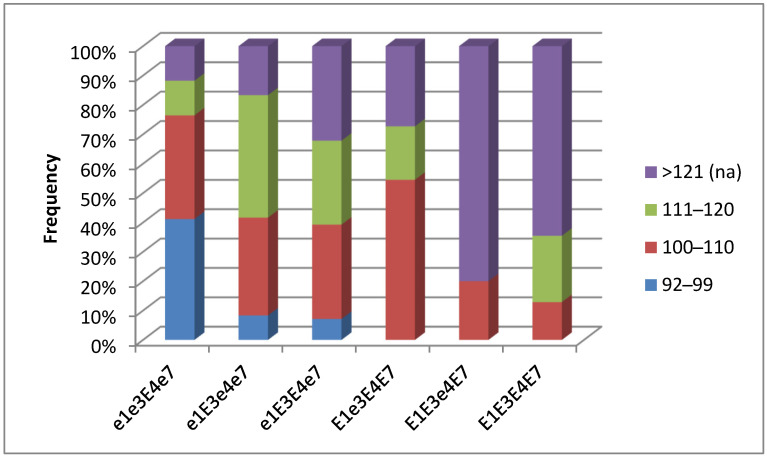
The frequency of occurrence of different periods (number of days) from seedling emergence to maturity in soybean genotypes from hybrid populations, F_4_ generation, grown in the Kostanay region, northern Kazakhstan.

**Figure 8 biomolecules-13-01146-f008:**
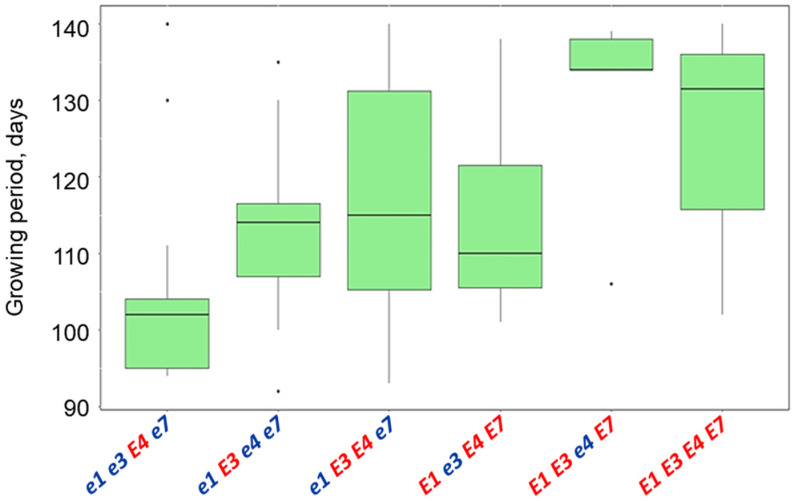
Interval plots of variance analysis for the growing duration of soybean breeding lines tested in the Kostanay region depending on their allelic composition of four *E* genes. Isolated dots represent extreme values exceeding the general variability level (0.99) of the received data. These dots are generated by R computer software and reflected automatically in the value of average and variability.

**Figure 9 biomolecules-13-01146-f009:**
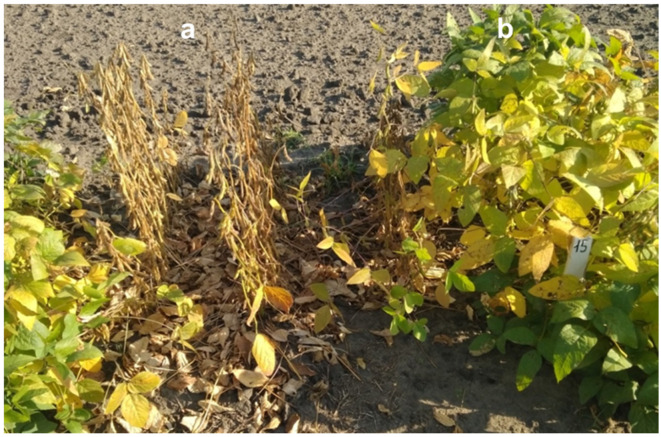
Images of soybean plants of two breeding lines with contrasting early and late maturing time at harvest, with genotypes *e1 e3 E4 e7* and *E1 e3 E4 E7*, indicated as (**a**) and (**b**), respectively. The lines originated from hybrid Birlik KV × Rana, generation F_4_, and were grown in doubled rows, 1 m in length each, in the Kostanay region, northern Kazakhstan, at high latitude.

**Table 1 biomolecules-13-01146-t001:** Soybean cultivars used in this study as parents for hybrid populations with their characteristics, both genotyping and phenotyping.

Name of Variety	Origin	M or F	Color of Flower	Maturity Group	*E1*	*E3*	*E4*	*E7*	Growth Type	Seed Yield in Plot (g)	TSW (g)
**A.** Maternal parents											
Birlik KV	Kazakhstan	♀	white	00	*e1*	*E3*	*E4*	*e7*	indetermin.	178.0 ^a^	163.6 ^b^
Zara	Kazakhstan	♀	white	I	*E1*	*E3*	*E4*	*E7*	determin.	195.1 ^a^	167.1 ^b^
**B.** Paternal parents crossed with cv. Birlik KV											
Rana	Czech	♂	purple	00	*E1*	*e3*	*E4*	*E7*	indetermin.	109.8 ^b^	145.1 ^b^
Toury	Czech	♂	purple	0	*e1*	*E3*	*E4*	*e7*	semi-determ.	170.9 ^a^	207.9 ^a^
Pripyat	Belarus	♂	purple	00	*e1*	*E3*	*e4*	*e7*	determin.	134.9 ^b^	163.5 ^b^
Grignon 5	France	♂	purple	I	*E1*	*E3*	*E4*	*E7*	indetermin.	110.8 ^b^	153.6 ^b^
Pamyat YGK	Kazakhstan	♂	purple	I	*E1*	*E3*	*E4*	*E7*	indetermin.	235.3 ^a^	180.5 ^a^
Soer 345	Russia	♂	purple	0	*E1*	*E3*	*E4*	*E7*	determin.	124.9 ^b^	152.7 ^b^
Altom	Russia	♂	purple	00	*e1*	*e3*	*e4*	*e7*	indetermin.	142.0 ^b^	217.0 ^a^
**C.** Paternal parents crossed with cv. Zara											
Bara	Russia	♂	purple	00	*E1*	*E3*	*e4*	*E7*	semi-determ.	162.5 ^a^	182.1 ^a^
Maleta	Russia	♂	purple	000	*e1*	*e3*	*e4*	*e7*	semi-determ.	94.1 ^b^	173.6 ^ab^
Soer 3	Russia	♂	purple	00	*e1*	*E3*	*E4*	*E7*	determin.	136.2 ^b^	162.2 ^b^
Ustya	Ukraine	♂	purple	00	*e1*	*E3*	*e4*	*e7*	determin.	134.1 ^b^	173.1 ^ab^
Yaselda	Belarus	♂	purple	00	*e1*	*E3*	*e4*	*e7*	indetermin.	123.2 ^b^	160.5 ^b^
Maple Amber	Canada	♂	purple	000	*e1*	*E3*	*E4*	*e7*	determin.	126.0 ^b^	203.1 ^a^
Fiskeby III	Sweden	♂	purple	00	*E1*	*e3*	*e4*	*e7*	determin.	139.3 ^b^	199.6 ^a^
Jhony	Unknown	♂	purple	00	*E1*	*E3*	*e4*	*e7*	indetermin.	135.7 ^b^	193.3 ^a^

Note: Values followed by the same letters are not significantly different at the *p* < 0.05 probability.

**Table 2 biomolecules-13-01146-t002:** Number of identified homozygote plants with recessive alleles of four *E* genes controlling photoperiod insensitivity in 11 soybean hybrids.

Hybrid Combination	Number of Identified Homozygote Plants with Recessive Alleles of Four *E* Genes				
	Number of F_2_–F_3_ Studied Plants	*e1e1*	*e3e3*	*e4e4*	*e7e7*
Birlik KV × Rana	70	28	40	- ^1^	28
Birlik KV × Pripyat	75	75	-	26	75
Birlik KV × Grignon 5	74	38	-	-	38
Birlik KV × Pamyat YGK	37	16	-	-	16
Birlik KV × Toury	77	77	-	-	77
Birlik KV × Soer 345	56	14	-	-	14
Zara × Bara	24	-	-	8	-
Zara × Maleta	146	48	17	22	48
Zara × Ustya	35	2	-	3	2
Zara × Soer 3	64	13	-	-	-
Zara × Jhony	9	-	-	6	0
Total	667	311	57	65	298

^1^ Note: Here and in other cases with mark ‘-’, the analysis was not conducted because both parents had dominant alleles in the *E* genes.

**Table 3 biomolecules-13-01146-t003:** Comparison of average values of the growing period of soybean from seedling emergence to flowering and to maturity in studied hybrid breeding lines, isolines, and local and international standards in the Kostanay region, northern Kazakhstan. Values followed by the same letters are not significantly different at the *p* < 0.05 probability.

Name and Genotype	Periods from Seedling Emergence to Flowering (Days)	Periods from Seedling Emergence to Maturity (Days)
Hybrid breeding lines		
*E1 E3 e4 E7*	49.4 ^a^	130.0 ^a^
*E1 E3 E4 E7*	46.7 ^a^	127.3 ^a^
*e1 E3 e4 e7*	44.3 ^ab^	116.7 ^ab^
*E1 e3 E4 E7*	42.7 ^ab^	114.0 ^ab^
*e1 E3 E4 e7*	42.0 ^ab^	113.9 ^ab^
*e* *1* *e* *3* *E* *4* *e* *7*	39.4 ^b^	104.0 ^b^
Isolines, and local and international standards		
Harosoy OT94-47 (*e1 e3 e4 e7*)	41 ^ab^	108 ^ab^
Harosoy OT89-5 (*e1 e3 e4 E7*)	43 ^ab^	110 ^ab^
Harosoy (*e1 E3 E4 E7*)	46 ^ab^	113 ^ab^
Maleta (*e1 e3 e4 E7*)	37 ^b^	98 ^b^
Ivushka (*E1 E3 E4 E7*)	39 ^b^	102 ^b^
SIBNIIK-315 (*e1 E3 e4 e7*)	39 ^b^	95 ^b^
Maple Amber (*e1 E3 E4 e7*)	38 ^b^	102 ^b^
Fiskeby III (*E1 e3 e4 e7)*	43 ^ab^	108 ^ab^

## Data Availability

The data presented in this manuscript are available from the corresponding authors upon request.

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
