# Peer review of "Marker-Assisted Selection for Early Maturing E Loci in Soybean Yielded Prospective Breeding Lines for High Latitudes of Northern Kazakhstan"

_biomolecules, 2023, doi:10.3390/biom13071146_

Round 1
Reviewer 1 Report
This manuscript reported a successful example for the application of molecular markers in identification and selection of early maturity of soybean in Northern Kazakhstan. The experience in developing superior performing ultra early-maturing soybean varieties can be shared by other countries with similar climates at high latitudes. The experiments were designed and conducted properly and the manuscript was well-prepared. However, some minor mistakes or questions should be corrected or answered.
L34: “World soybean production is growing every year” is not always correct.
L66-67: The description of “Dominant alleles of E9 were reported as controlling early flowering and maturity [22-23], with a similar effect to the dominant alleles of the E6 gene” is not accurate because E9 and E7 are different genes (mutated GmFT2a and J).
L73-74: “soybean genotypes with dominant alleles of J are very suitable for lower rather than higher latitudes” is wrong. In fact, the mutated or recessive J (j) confers the long juvenility of tropical soybean (Yue et al., 2017; Lu et al., 2017). References are below:
Yue Y, Liu N, Jiang B, Li M, Wang H, Jiang Z, Pan H, Xia Q, Ma Q, Han T, Nian H (2017) A single nucleotide deletion in J encoding GmELF3 confers long juvenility and is associated with adaption of tropic soybean. Mol Plant 10:656-658. https://doi.org/10.1016/j.molp.2016.12.004
Lu S, Zhao X, Hu Y, Liu S, Nan H, Li X, Fang C, Cao D, Shi X, Kong L, Su T, Zhang F, Li S, Wang Z, Yuan X, Cober ER, Weller JL, Liu B, Hou X, Tian Z, Kong F (2017) Natural variation at the
soybean J locus improves adaptation to the tropics and enhances yield. Nat Genet 49:773-779.
https://doi.org/10.1038/ng.3819
L191-192: “hybrid soybean plants” is easy to be misunderstood as hybrid for heterosis utilization in F1, please replace it with other words juch as progenies.
L255-256: Please provide the references for International Standard (cultivars). Does it mean the E isolines developed by Bernard et al. in 1970s and 1980s?
L294-300: In Fig.1b, Line with E1E1 (cv. ОТ93-26, 3rd from the right) has two bands, how to explain it?
L386-390: In Fig.4, please illustrate the relationship between the round-shaped color and association.
L398-399: “the transition to flowering” is not a correct expression. I suggest the authors to replace it with “days to flowering after emergence” or “flowering time”.
L4510-412: Should “germination” be “emergence”? As I understand, germination is not easy to observed because the seeds (even germinated) were in the soil before emergence.
L418-422: Please explain the reason why the genotype of E1E3e4E7 flowered later than E1E3E4E7 in the section of Discussion.
Line 423-424: “Comparison of the studied breeding lines to control lines Harosoy” should be between the same genotype of E loci. Please change the expression.
L428: “growing season” usually means the frost-free period for growth of plant. Please replace “season” with “period”.
Table 3: “International standards” is not a standardized expression. In fact, they were isogenic lines (isolines).
L447: “growing season duration” should be “growing duration” or “growth period”. The expression should be uniform in the manuscript.
L448: “field trial tests” should be “field trials” or “field tests”.
L451: “vegetative period” should be “growth period”. Vegetative period should be before the reproductive (post-flowering) period.
L463: “effect of temperature” may be important but there was no evidence to support the hypothesis in the current study.
Fig. 8: The genotypes of two lines should be provided.
L476-477: Do the authors mean “insensitivity of plants to longer day-length”? Does “vegetative period” mean “growth period” here?
L553: “growing season” can be changed to “growth period”.
L589, L602, L641, L671: “PLoS one” should be “PLoS ONE”.
The manusctript was well written in English.
Author Response
Reviewer 1:
This manuscript reported a successful example for the application of molecular markers in identification and selection of early maturity of soybean in Northern Kazakhstan. The experience in developing superior performing ultra early-maturing soybean varieties can be shared by other countries with similar climates at high latitudes. The experiments were designed and conducted properly and the manuscript was well-prepared. However, some minor mistakes or questions should be corrected or answered.
(1) L34: “World soybean production is growing every year” is not always correct.
Response: The statement was corrected.
(2) L66-67: The description of “Dominant alleles of E9 were reported as controlling early flowering and maturity [22-23], with a similar effect to the dominant alleles of the E6 gene” is not accurate because E9 and E7 are different genes (mutated GmFT2a and J).
Response: The fragment was modified for clarity. However, we did not claim that E9 gene affected E6 gene. In contrast, dominant alleles of both genes, E9 and E6, had similar phenotyping effect for early flowering and maturity. There is no link to E7 gene in this context.
(3) L73-74: “soybean genotypes with dominant alleles of J are very suitable for lower rather than higher latitudes” is wrong. In fact, the mutated or recessive J (j) confers the long juvenility of tropical soybean (Yue et al., 2017; Lu et al., 2017). References are below:
Yue Y, Liu N, Jiang B, Li M, Wang H, Jiang Z, Pan H, Xia Q, Ma Q, Han T, Nian H (2017) A single nucleotide deletion in J encoding GmELF3 confers long juvenility and is associated with adaption of tropic soybean. Mol Plant 10:656-658. https://doi.org/10.1016/j.molp.2016.12.004
Lu S, Zhao X, Hu Y, Liu S, Nan H, Li X, Fang C, Cao D, Shi X, Kong L, Su T, Zhang F, Li S, Wang Z, Yuan X, Cober ER, Weller JL, Liu B, Hou X, Tian Z, Kong F (2017) Natural variation at the soybean J locus improves adaptation to the tropics and enhances yield. Nat Genet 49:773-779. https://doi.org/10.1038/ng.3819
Response: The fragment was modified and additional sentence was inserted together with two suggested references.
(4) L191-192: “hybrid soybean plants” is easy to be misunderstood as hybrid for heterosis utilization in F1, please replace it with other words juch as progenies.
Response: Corrected.
(5) L255-256: Please provide the references for International Standard (cultivars). Does it mean the E isolines developed by Bernard et al. in 1970s and 1980s?
Response: References for International standards were added. Both local and International standards have no any relationship with Harosoy and isolines and used for different purposes.
(6) L294-300: In Fig.1b, Line with E1E1 (cv. ОТ93-26, 3rd from the right) has two bands, how to explain it?
Response: Sometimes, the use of SSR markers can cause unspecific amplification in some genotypes, as it happened here with use of Satt557. The reason of this phenomenon remains unknown but it does not affect the marker scoring for plant genotyping.
(7) L386-390: In Fig.4, please illustrate the relationship between the round-shaped color and association.
Response: The fragment was modified and explanation was added.
(8) L398-399: “the transition to flowering” is not a correct expression. I suggest the authors to replace it with “days to flowering after emergence” or “flowering time”.
Response: The phrase was modified.
(9) L410-412: Should “germination” be “emergence”? As I understand, germination is not easy to observed because the seeds (even germinated) were in the soil before emergence.
Response: Corrected.
(10) L418-422: Please explain the reason why the genotype of E1E3e4E7 flowered later than E1E3E4E7 in the section of Discussion.
Response: Mechanisms of functions and interactions E loci are complicated with possible involvement of other genes. However, the difference for average flowering and maturity time between these two genotypes accounted for 2.7 days and was statistically non-significant. The sentence was inserted in the text.
(11) Line 423-424: “Comparison of the studied breeding lines to control lines Harosoy” should be between the same genotype of E loci. Please change the expression.
Response: The sentence was modified.
(12) L428: “growing season” usually means the frost-free period for growth of plant. Please replace “season” with “period”.
Response: Corrected.
(13) Table 3: “International standards” is not a standardized expression. In fact, they were isogenic lines (isolines).
Response: In the Table 3, ‘controls’ were changed for ‘isolines’ as suggested. Both International and local standards are used in field trials for agronomic comparison with studied genotypes. In some countries, such ‘standards’ called ‘checks’ with the same meaning as synonyms. Our comment about International standard is also mentioned present in Point 5 above and description inserted in M&M section..
(14) L447: “growing season duration” should be “growing duration” or “growth period”. The expression should be uniform in the manuscript.
Response: Corrected in entire manuscript.
(15) L448: “field trial tests” should be “field trials” or “field tests”.
Response: Modified.
(16) L451: “vegetative period” should be “growth period”. Vegetative period should be before the reproductive (post-flowering) period.
Response: Corrected.
(17) L463: “effect of temperature” may be important but there was no evidence to support the hypothesis in the current study.
Response: The phrase was added.
(18) Fig. 8: The genotypes of two lines should be provided.
Response: Genotype identification was added.
(19) L476-477: Do the authors mean “insensitivity of plants to longer day-length”? Does “vegetative period” mean “growth period” here?
Response: Corrected.
(20) L553: “growing season” can be changed to “growth period”.
Response: Corrected.
(21) L589, L602, L641, L671: “PLoS one” should be “PLoS ONE”.
Response: Corrected.

Reviewer 2 Report
This manuscript addresses maturity genes in a survey of soybean genotypes adapted primarily to a specific region and a follow up use of molecular markers to select for maturity adaptation within segregating populations. The manuscript is well written and clearly outlines the work and results of the work. I have a few comments and suggestions on the attached file. Note that there are a series of alleles at the E1 locus with E1 > e1-as > el-nl or e1-fs (for maturity delay). Can this work distinguish between e1-as and the fully null e1 alleles?

Author Response
Reviewer 2:
This manuscript addresses maturity genes in a survey of soybean genotypes adapted primarily to a specific region and a follow up use of molecular markers to select for maturity adaptation within segregating populations. The manuscript is well written and clearly outlines the work and results of the work. I have a few comments and suggestions on the attached file. Note that there are a series of alleles at the E1 locus with E1 > e1-as > el-nl or e1-fs (for maturity delay). Can this work distinguish between e1-as and the fully null e1 alleles?
(1) L2. Title: E loci
Response: Corrected.
(2) L61. The E5 locus was demonstrated not to exist and likely confused with the E2 locus in this paper Breeding Science 66: 407–415 (2016)
doi:10.1270/jsbbs.15160. https://www.jstage.jst.go.jp/article/jsbbs/66/3/66_15160/_article
Response: The sentence with reference was added.
(3) L131. Maple Amber but not Mapleamber
Response: Corrected.
(4) L267. Insert ‘to’ in the phrase: ‘Prior the start…’
Response: Corrected.
(5) L279-294. Note that there are a series of alleles at E1 with E1 > e1-as > el-nl or e1-fs (for maturity delay). Can the work distinguish between e1-as and the fully null e1 alleles?
Response: A paragraph was added in the Discussion section addressing this point.
In our studies, we used SSR markers that distinguish only dominant (E1) and recessive (e1) alleles according to Molnar S.J., 2003.
(6) L409-412. Figure 5: Could this purple bar be labelled >121
Response: The label was added.
(7) L451. ‘Growing period’ but not ‘vegetative period’.
Response: Corrected.
(8) L504-508. This paper also mentions the minor role of e4e4: https://www.frontiersin.org/articles/10.3389/fpls.2018.01286/full
Response: A sentence and reference was added.

Reviewer 3 Report
Dear Authors,
I have reviewed your manuscript "Marker-assisted selection for early-maturing loci E in soybean [Glycine max (L.) Merr.] resulted in prospective breeding lines for high latitudes in Northern Kazakhstan", submitted for publication in Biomolecules.
Reading your manuscript I was impressed by your work, by both the quantity and quality of your results, and by the way in which you presented them. I was also happy to find out that your research has a tangible potential to contribute to the improvement of soybean production in your country. For that matter I want to congratulate you.
However, in order to make your paper even better than it already is, I have compiled a list of suggestions for improvement. Certain clarifications are needed, most of all because in the multitude of the experiments that you have carried out and of the results that you presented, the reader gets easily lost and certain important points need to be better stressed so that your work can be better understood. My remarks may appear numerous, but most of them can be considered minor and I believe that you will easily address them, please see below:
· Experimental design, major (general) remarks:
o Supplementary Material is cited at several points throughout the paper, but I could not find it within your submission.
o It is not very clear from your paper why have some of your field trials been carried out in the South of the country. The major focus of your research is on Northern Kazakhstan, and all of your important field trials have been done in the North; however, some of your field experiments have been carried out in the South as well. You do provide some explanation in the Material & Methods section (part 2.4, lines 182-190), but two points remain insufficiently explained, or at least, the reader cannot readily grasp them from the context, so that a more explicit explanation is needed, both in the M&M section, and especially in the Results section:
§ 1) Which experiments have been carried out in the North, and which ones in the South of the country, and why? (What is the rationale?) Please thoroughly elaborate within the manuscript.
§ 2) Which results (in the Results section) have been obtained from the field trials in the North, and which ones in the South? Are there results (such as seed yield, or TGW, for example in Table 1) which have been collected both from the fields in the North and in the South? If yes, please show them separately, and maybe also provide some commentary regarding the observed differences.
o The twenty early maturing genotypes that you identified as promising for cultivation in Northern Kazakhstan (lines 458-460) are the most important outcome of your paper. However, their agronomical output in terms of seed yield or TGW is not presented within your paper, or at least not within the main manuscript. If you have determined these traits (or any other agronomically important traits related to early maturation), it would be essential to present these results within your main manuscript, and to compare them to the same traits in the late-maturing genotypes. Such result would make your research more complete and its outcome more tangible.
o The names of the parental genotypes are not uniformly spelled throughout the manuscript. I have found both "Birlik" and "Burlik", both "Soer 345" and "Coer 345", both "Pamyat YKG" and "Memory YKG" at different points within the manuscript. Please double-check all the occurencies of all the genotype names and thoroughly correct them throughout the manuscript text.
· Manuscript title: I find that your manuscript title is too long and complicated. I would consider removing the Latin name, as Glycine max is a very dominantly cultivated species. (On that note, "Merr." should not be written in italic.) Also, my personal opinion is that "resulted in" does not sound very good in a manuscript title. Anyway, I gave you a couple of ideas, and you should decide what to do with them.
· Abstract – line 23: were identified
· Introduction:
o line 32: I would say "one of the world's leading oilseed crops". Is it produced more than oilseed rape, and if so, is it produced for oil more than oilseed rape? If you are sure that soybean is really the leading oilseed crop, you should support that claim by immediately citing FAOSTAT (which is currently ref.no.2).
o line 58 and 68: please replace "where" with "whereby"
o line 75: please replace "word" with "world"
o line 79-80: I would say that "the e7 allele was also indicated as insensitive to photoperiod". Please double-check.
· Materials & Methods:
o lines 125-128: "only one dominant allele", "only one recessive allele", etc. Regarding Harosoy OT89-5, Harosoy, and Harosoy OT93-26, I believe that here, we are talking about diploid genotypes which are homozygotized at the loci E1, E3, E4, E7. (Did I get it right that they are homozygotized?) If so, you cannot say that "only one allele" is dominant or recessive, but that only one of the concerned loci contains dominant (or recessive) alleles, whereby both alleles of the concerned genes are in their dominant (or recessive) form because these genotypes are homozygous for all of them. I know it is more complicated to formulate this way, but it is important to be clear here.
o line 170: I would ask the Authors that they add all the primers that they used in their PCR analyses, together with their primer accession numbers, if they are not already provided within the Supplementary Material. Please ignore this comment if you have already done that, but as I said, I did not have access to your Supplementary Material, so I could not see whether the primers are there.
o line 190: the East coordinate (76°54' E) of the breeding station "Zarechnoe" is incorrect and has been copy-pasted from the above coordinates of Almaty. Looking at a map of Kazakhstan it is obvious that the Kostanay region lies much further West of that.
o lines 182-190: Please add the basic climatological data for both regions (climate type according to the Köppen classification, mean annual temperature and rainfall).
· Results:
o line 235-236: Please add an explanation. The readers who are not plant breeders might not see the connection between flower color and the size of unopened flowers.
o Table 1: Please see my above comment regarding which results were obtained from the North and which ones from the South of the country.
o line 257-263: The results of hybridization efficiency should be provided in a table, at least within Supplementary Material if not in the main manuscript. The appropriate Table should then by cited accordingly, here in the text of the Results.
o line 268: This sentence should be divided into two sentences. The first sentence ends with "(Table 1)", and then there is a second sentence, whose beginning, referring to the paternal lines, is currently missing from the text.
o line 320: I believe that, according to Mendel, they did not have only dominant alleles; they had dominant alleles, which was sufficient to exclude them, because a single dominant allele is sufficient to confer the dominant trait to the concerned genotype.
o line 328: "as well as and positive and negative controls" – please revise
o line 422: P should be written in italic.
o Table 3, header (right-hand side): please replace "mature" with "maturity"
o line 431-440: this part of the text should be transferred elsewhere, to the point where Figure 5 was introduced, previously. So that the results shown in the Figures are narrated in the proper chronological order.
o line 464: 26 out of the 35 E1 E3 E4 E7 lines, I suppose?
· Discussion:
o line 482: please replace "soybean cultivars were produced" with "soybean cultivars ARE produced"
o line 486, 488, 547: please replace "from" with "out of"
o line 490: please replace "where" with "whereby"
o line 497-499: In your work, you did not seem to obtain a single genetic line with all the four loci in homozygous-recessive state (e1e1 e3e3 e4e4 e7e7). Is there a particular reason for this, or was it just matter of "stochastic luck"? A brief commentary about this should be mandatory within the Discussion section, since the entire Introduction of your manuscript suggests that a genotype with all these recessive alleles would be likely to perform fantastically well in terms of early maturation.
o line 513: E4, or E3? I am not sure from the context, please double-check.
o line 530: You do not need to repeat "Northern Kazakhstan" after "Kostanay region" throughout the entire manuscript. At the beginning of the paper you indicated where Kostanay is, so you do not have to repeat it all the way. It represents a burden to the text when it is repeated so many times.
I look forward to reading your revised paper in its published form in Biomolecules.
Kind regards,
Reviewer

The quality of the English language is mostly very well, however I have found several points within the manuscript where minor corrections are required. However, since they were not numerous and were simple to correct, I indicated the appropriate corrections within my Comments to the Authors.
Author Response
Reviewer 3:
Dear Authors,
I have reviewed your manuscript "Marker-assisted selection for early-maturing loci E in soybean [Glycine max (L.) Merr.] resulted in prospective breeding lines for high latitudes in Northern Kazakhstan", submitted for publication in Biomolecules.
Reading your manuscript I was impressed by your work, by both the quantity and quality of your results, and by the way in which you presented them. I was also happy to find out that your research has a tangible potential to contribute to the improvement of soybean production in your country. For that matter I want to congratulate you.
However, in order to make your paper even better than it already is, I have compiled a list of suggestions for improvement. Certain clarifications are needed, most of all because in the multitude of the experiments that you have carried out and of the results that you presented, the reader gets easily lost and certain important points need to be better stressed so that your work can be better understood. My remarks may appear numerous, but most of them can be considered minor and I believe that you will easily address them, please see below:
Experimental design, major (general) remarks:
(1) Supplementary Material is cited at several points throughout the paper, but I could not find it within your submission.
Response: Supplementary materials were added to the manuscript now.
(2) It is not very clear from your paper why have some of your field trials been carried out in the South of the country. The major focus of your research is on Northern Kazakhstan, and all of your important field trials have been done in the North; however, some of your field experiments have been carried out in the South as well. You do provide some explanation in the Material & Methods section (part 2.4, lines 182-190), but two points remain insufficiently explained, or at least, the reader cannot readily grasp them from the context, so that a more explicit explanation is needed, both in the M&M section, and especially in the Results section
Response: We provided our responses in more detail in the following points below.
(3) Which experiments have been carried out in the North, and which ones in the South of the country, and why? (What is the rationale?) Please thoroughly elaborate within the manuscript.
Response: The description and details of experiments carried out in Southern and Norther Kazakhstan were added in M&M, Results and Introduction sections.
(4) Which results (in the Results section) have been obtained from the field trials in the North, and which ones in the South? Are there results (such as seed yield, or TGW, for example in Table 1) which have been collected both from the fields in the North and in the South? If yes, please show them separately, and maybe also provide some commentary regarding the observed differences.
Response: The description and details of experiments carried out in Southern and Norther Kazakhstan were added in M&M, Results and Introduction sections.
(5) The twenty early maturing genotypes that you identified as promising for cultivation in Northern Kazakhstan (lines 458-460) are the most important outcome of your paper. However, their agronomical output in terms of seed yield or TGW is not presented within your paper, or at least not within the main manuscript. If you have determined these traits (or any other agronomically important traits related to early maturation), it would be essential to present these results within your main manuscript, and to compare them to the same traits in the late-maturing genotypes. Such result would make your research more complete and its outcome more tangible.
Response: The brief description of this material; was added in the text and complete data are present in Supplementary material S5.
(6) The names of the parental genotypes are not uniformly spelled throughout the manuscript. I have found both "Birlik" and "Burlik", both "Soer 345" and "Coer 345", both "Pamyat YKG" and "Memory YKG" at different points within the manuscript. Please double-check all the occurencies of all the genotype names and thoroughly correct them throughout the manuscript text.
Response: Corrected and checked.
(7) Manuscript title: I find that your manuscript title is too long and complicated. I would consider removing the Latin name, as Glycine max is a very dominantly cultivated species. (On that note, "Merr." should not be written in italic.) Also, my personal opinion is that "resulted in" does not sound very good in a manuscript title. Anyway, I gave you a couple of ideas, and you should decide what to do with them.
Response:: The Title was modified as we believe the best way to reflect the content of our manuscript.
(8) Abstract – line 23: were identified
Response: Corrected.
(9) Introduction. line 32: I would say "one of the world's leading oilseed crops". Is it produced more than oilseed rape, and if so, is it produced for oil more than oilseed rape? If you are sure that soybean is really the leading oilseed crop, you should support that claim by immediately citing FAOSTAT (which is currently ref.no.2).
Response: More information and reference was added in the fragment.
(10) line 58 and 68: please replace "where" with "whereby"
Response: Corrected.
(11) line 75: please replace "word" with "world"
Response: Corrected.
(12) line 79-80: I would say that "the e7 allele was also indicated as insensitive to photoperiod". Please double-check.
Response: Corrected.
Materials & Methods:
(13) lines 125-128: "only one dominant allele", "only one recessive allele", etc. Regarding Harosoy OT89-5, Harosoy, and Harosoy OT93-26, I believe that here, we are talking about diploid genotypes which are homozygotized at the loci E1, E3, E4, E7. (Did I get it right that they are homozygotized?) If so, you cannot say that "only one allele" is dominant or recessive, but that only one of the concerned loci contains dominant (or recessive) alleles, whereby both alleles of the concerned genes are in their dominant (or recessive) form because these genotypes are homozygous for all of them. I know it is more complicated to formulate this way, but it is important to be clear here.
Response:: Modified as suggested.
(14) line 170: I would ask the Authors that they add all the primers that they used in their PCR analyses, together with their primer accession numbers, if they are not already provided within the Supplementary Material. Please ignore this comment if you have already done that, but as I said, I did not have access to your Supplementary Material, so I could not see whether the primers are there.
Response: Supplementary material S1 was included to the manuscript now.
(15) line 190: the East coordinate (76°54' E) of the breeding station "Zarechnoe" is incorrect and has been copy-pasted from the above coordinates of Almaty. Looking at a map of Kazakhstan it is obvious that the Kostanay region lies much further West of that.
Response: Modified.
(16) lines 182-190: Please add the basic climatological data for both regions (climate type according to the Köppen classification, mean annual temperature and rainfall).
Response: The information was added.
Results:
(17) line 235-236: Please add an explanation. The readers who are not plant breeders might not see the connection between flower color and the size of unopened flowers.
Response: This fragment was re-written completely.
(18) Table 1: Please see my above comment regarding which results were obtained from the North and which ones from the South of the country.
Response: This point was addressed earlier in Points 3-4.
(19) line 257-263: The results of hybridization efficiency should be provided in a table, at least within Supplementary Material if not in the main manuscript. The appropriate Table should then by cited accordingly, here in the text of the Results.
Response: This information was added in the text and in additional Supplementary material S2.
(20) line 268: This sentence should be divided into two sentences. The first sentence ends with "(Table 1)", and then there is a second sentence, whose beginning, referring to the paternal lines, is currently missing from the text.
Response: This sentence was split for two modified. The second sentence now is describing all parental cultivars used in this study.
(21) line 320: I believe that, according to Mendel, they did not have only dominant alleles; they had dominant alleles, which was sufficient to exclude them, because a single dominant allele is sufficient to confer the dominant trait to the concerned genotype.
Response: The fragment was modified and additional sentence was inserted.
(22) line 328: "as well as and positive and negative controls" – please revise
Response: We re-phrased this fragment in the legends of Figures 1-3, trying to address this point of the Reviewer.
(23) line 422: P should be written in italic.
Response:: Corrected.
(24) Table 3, header (right-hand side): please replace "mature" with "maturity"
Response: Corrected.
(25) line 431-440: this part of the text should be transferred elsewhere, to the point where Figure 5 was introduced, previously. So that the results shown in the Figures are narrated in the proper chronological order.
Response: We understood the Reviewer’s point but decided to split Figure 5, parts ‘a’ and ‘b’, into two Figures. Therefore, new Figure 7 (former Figure 5b) is now following after this fragment about time to maturity.
(26) line 464: 26 out of the 35 E1 E3 E4 E7 lines, I suppose?
Response: One sentence was added in the text addressing this point.
Discussion:
(27) line 482: please replace "soybean cultivars were produced" with "soybean cultivars ARE produced"
Response: Corrected.
(28) line 486, 488, 547: please replace "from" with "out of"
Response: Corrected.
(29) line 490: please replace "where" with "whereby"
Response: Corrected.
(30) line 497-499: In your work, you did not seem to obtain a single genetic line with all the four loci in homozygous-recessive state (e1e1 e3e3 e4e4 e7e7). Is there a particular reason for this, or was it just matter of "stochastic luck"? A brief commentary about this should be mandatory within the Discussion section, since the entire Introduction of your manuscript suggests that a genotype with all these recessive alleles would be likely to perform fantastically well in terms of early maturation.
Response:: In our experiments, genotypes e1e1 e3e3 e4e4 e7e7 were not found. One of the possible reasons that prior the genotyping, plants were go thought preliminary selection and eliminated for disease infections, dwarfism, lodging and dehiscence. Therefore, it is possible that plants with homozygous-recessive alleles in four studied E loci were eliminated before the genotyping.
(31) line 513: E4, or E3? I am not sure from the context, please double-check.
Response: The gene was E4 and the phrase was modified for the clarity.
(32) line 530: You do not need to repeat "Northern Kazakhstan" after "Kostanay region" throughout the entire manuscript. At the beginning of the paper you indicated where Kostanay is, so you do not have to repeat it all the way. It represents a burden to the text when it is repeated so many times.
Response: Corrected in this and some other cases in the text. However, we have to point out to the Reviewer that we cannot delete ‘Northern Kazakhstan’ from Figure legends and Table captions. This is because each Figure and Table must be self-explanatory and understandable without references to the text. Therefore, ‘Kostanay region, Northern Kazakhstan’ must be present in Figures and Tables as it now.
